# Acetylcholine Sustains LNCaP Prostate Cancer Cell Migration, Invasion and Proliferation Through Glyoxalase 1/MG-H1 Axis with the Involvement of Osteopontin

**DOI:** 10.3390/ijms26094107

**Published:** 2025-04-25

**Authors:** Dominga Manfredelli, Tatiana Armeni, Lidia de Bari, Andrea Scirè, Vincenzo Nicola Talesa, Cinzia Antognelli, Marilena Pariano

**Affiliations:** 1Department of Medicine and Surgery, Università Degli Studi di Perugia, 06129 Perugia, Italy; dominga.manfredelli@dottorandi.unipg.it (D.M.); vincenzo.talesa@unipg.it (V.N.T.); marilena.pariano@unipg.it (M.P.); 2Department of Life and Environmental Sciences (Di.S.V.A.), Università Politecnica delle Marche, 60131 Ancona, Italy; t.armeni@staff.univpm.it; 3Institute of Biomembranes, Bioenergetics and Molecular Biotechnologies (IBIOM), National Research Council (CNR), 70126 Bari, Italy; l.debari@ibiom.cnr.it; 4Department of Odontostomatologic and Specialized Clinical Sciences, Università Politecnica delle Marche, 60131 Ancona, Italy; a.a.scire@staff.unipvm.it

**Keywords:** acetylcholine, Glyoxalase 1, osteopontin, MG-H1, prostate cancer, LNCaP, migration, invasion

## Abstract

The neurotransmitter acetylcholine (ACh) plays a pro-carcinogenic role in various cancer types, including prostate cancer (PCa). The existing body of knowledge concerning the mechanisms that underpin the protumoral role of ACh in PCa is limited. Glyoxalase 1 (Glo1) is a metabolic enzyme that removes methylglyoxal (MG), an endogenous post-translational modification agent, generating 5-hydro-5-methylimidazolone (MG-H1). The Glo1/MG-H1 axis is involved in PCa tumorigenesis and progression. By using LNCaP and PC3 PCa cells, representing extensively studied cell models of poorly aggressive and bone metastasis-derived PCa, respectively, we found that ACh specifically sustains LNCaP cell migration, invasion and proliferation through Glo1-dependent MG-H1 accumulation with the involvement of osteopontin (OPN), thus providing a novel mechanism underlying ACh’s protumoral role in PCa cells. The findings of this study unveil a hitherto unidentified mechanism implicated in the progression of PCa, which is initiated by ACh and involves both the Glo1/MG-H1 axis and OPN. This discovery provides the basis for new avenues of in vivo investigation into the physiological relevance of the roles of the ACh-driven Glo1/MG-H1 axis and OPN in PCa progression and for further research aimed at exploring new ways of managing PCa progression, with the aim of preventing the disease from becoming incurable.

## 1. Introduction

The nervous system plays a crucial role in cancer initiation and progression as part of the tumor microenvironment [1]. In particular, emerging evidence indicates that the sympathetic nervous system with its main neurotransmitter acetylcholine (ACh) promotes carcinogenesis through the control of a number of biological responses, including cell proliferation, migration and angiogenesis [2,3]. In prostate cancer (PCa), ACh contributes to sustaining cancer progression by promoting cell proliferation, migration and invasion [3,4] through various receptor-dependent mechanisms [5]. For instance, in 2015, Wang and colleagues [4] described that the autocrine triggering of ACh-activated cholinergic muscarinic receptor 1 (CHRM1) promotes PC3 cell growth via CaM/CaMKK-mediated phosphorylation of Akt. Three years later, Yin and collaborators reported that the same CHRM1 was involved in regulating PC3 and LNCaP PCa cell migration and invasion through hedgehog signaling pathway activation [5]. More recently, it has been described that nicotinic ACh receptor subunit α5 (α5-nAChR) is involved in the proliferation and invasion of DU145 and PC3 cell lines [6]. Despite these advancements, knowledge of the mechanisms underlying the protumoral role of ACh in PCa remains very poor. Further insight into the knowledge of these mechanisms may offer promising opportunities for additional studies aimed at developing novel strategies for the control of PCa genesis and progression.

Glyoxalase 1 (Glo1) is a GSH-dependent enzyme whose major function is the metabolization of methylglyoxal (MG), a highly reactive α-dicarbonyl compound acting as a potent post-translational modification agent, through a process known as glycation [7]. In this role, MG preferentially binds to arginine residues of proteins, generating 5-hydro-5-methylimidazolone (MG-H1), which belongs to the heterogeneous family of compounds named advanced glycation end products (AGEs). In PCa, both MG, through MG-H1, and Glo1 have been involved in the promotion of invasion and migration, thus acting as a pivotal player in PCa progression [8,9].

Osteopontin (OPN) is a multifunctional glycoprotein playing a crucial oncogenic role by conferring to cancer cells a migratory phenotype and driving signaling pathways that induce proliferation, invasion and metastasis [10,11], including in PCa [12,13,14]. Multiple molecular mechanisms, including integrin signaling, NF-κB and PI3K/Akt activation, and interactions with CD44 are involved in OPN-triggered tumorigenesis [15].

Based on these premises, in the present study, we investigated whether the Glo1/MG-H1 axis was involved in ACh’s protumorigenic role through OPN control in two human PCa cell lines, LNCaP and PC3, representing extensively studied cell models of poorly aggressive [16,17] and bone metastasis-derived PCa [16,17], respectively. We found that ACh sustains LNCaP and PC3 PCa cell migration, invasion and proliferation through separate mechanisms of action, and we give evidence of a novel mechanism based on Glo1-dependent MG-H1-mediated OPN upregulation in LNCaP cells.

Altogether, our findings reveal a novel mechanism triggered by ACh involved in PCa progression, thus paving the way for additional investigations designed to explore new strategies for the control of PCa progression, notably, before it becomes clinically incurable.

## 2. Results

### 2.1. Effect of ACh on LNCaP and PC3 Cell Migration and Invasion

LNCaP and PC3 cells were treated with ACh at different concentrations (0 µM, 0.1 µM, 0.5 µM, 1 µM, 5 µM and 10 µM) for 6, 24 and 48 h, and cell migration and invasion were evaluated. As shown in Figure 1, cell migration (Figure 1a) and invasion (Figure 1b) significantly increased in LNCaP cells at the concentrations of 5 µM and 10 µM after 24 and 48 h.

Cell migration (Figure 2a) and invasion (Figure 2b) also significantly increased in PC3 cells at the concentrations of 5 µM and 10 µM after 24 and 48 h.

### 2.2. Effect of ACh on LNCaP and PC3 Cell Proliferation

It is known that ACh contributes to sustaining cancer progression in PCa by promoting cell proliferation as well [3,4]. Here, we confirm the pro-proliferative effect of ACh on both LNCaP and PC3 cells at 5 µM and 10 µM after 24 and 48 h (Figure 3).

Since 5 µM ACh 24 h post-exposure already exhibited a significant effect on both migration and invasion as well as on proliferation in both cell lines, this experimental condition was chosen for the subsequent experiments.

### 2.3. Effect of ACh on Glo1 Expression and MG-H1 Levels in LNCaP and PC3 Cells

To investigate whether Glo1 and MG-H1 were involved in ACh-induced migration, invasion and proliferation, both LNCaP and PC3 cells were treated with 5 µM ACh for 24 h. As shown in Figure 4a, in LNCaP cells, Glo1 transcript levels significantly decreased compared to untreated cells. A similar trend was observed for Glo1 protein expression (Figure 4a) as well as for Glo1-specific activity (Figure 4a). Accordingly, MG-H1 levels markedly increased upon the addition of ACh compared to control cells (Figure 4a). Conversely, neither Glo1 expression nor MG-H1 levels were affected by ACh treatment in PC3 cells (Figure 4b). These results suggest that the pro-migrating, pro-invasive and pro-proliferative effects of ACh, associated with Glo1 down-regulation and the consequent MG-H1 accumulation, are specific to poorly aggressive LNCaP cells.

### 2.4. Effects of ACh on Osteopontin (OPN) Expression in LNCaP and PC3 Cells

To investigate whether OPN, a key multifaceted regulator in tumor progression [18], was involved in ACh-induced migration and invasion, both LNCaP and PC3 cells were treated with 5 µM ACh for 24 h. As shown in Figure 5a, in LNCaP cells, ACh induced a significant increase in both OPN mRNA and protein levels compared to untreated cells. Conversely, only a modest, not significant, increase was observed in PC3 cells (Figure 5b).

These results suggest that the pro-migrating, pro-invasive and pro-proliferating effects of ACh are associated with OPN up-regulation and that, again, it is specific only to poorly aggressive LNCaP cells.

Therefore, subsequent experiments aimed at investigating if ACh-driven migration, invasion and proliferation causatively involved the Glo1/MG-H1 axis and OPN were conducted only in LNCaP cells.

### 2.5. ACh Sustains LNCaP Migration, Invasion and Proliferation Through Glo1-Dependent MG-H1-Mediated Osteopontin Upregulation

In order to demonstrate that ACh drove migration, invasion and proliferation via the Glo1/MG-H1 axis and OPN, LNCaP cells were pre-treated with carnosine, a potent MG scavenger [19], thus a reducer of MG-derived MG-H1 [20]. As shown in Figure 6, 10 mM carnosine pre-treatment for 48 h abrogated ACh-induced OPN up-regulation and restored all the biological responses considered. These results robustly support the hypothesized mechanism by which ACh sustains LNCaP PCa cell migration, invasion and proliferation: Glo1-dependent MG-H1-mediated osteopontin upregulation.

### 2.6. Pre-Treatment of LNCaP PCa Cells with Donepezil, an Acetylcholinesterase (AChE) Inhibitor, and ACh Further Supports the Involvement of the Glo1/MG-H1 Axis and Osteopontin in the Pro-Tumorigenic Effect Driven by ACh Through Cell Migration, Invasion and Proliferation Control

To further support the involvement of the Glo1/MG-H1 axis and osteopontin as part of the pro-tumorigenic effect driven by ACh through cell migration, invasion and proliferation control, LNCaP cells were exposed to both donepezil (DNPZ)—a selective inhibitor of Acetylcholinesterase (AChE) [21], the major enzyme metabolizing ACh [22]—and ACh. After verifying DNPZ’s inhibitory efficacy on AChE activity by both spectrophotometric (Figure 7a) and electrophoresis methods (Figure 7b), as well as on ACh levels (Appendix A), we found that, upon ACh treatment, DNPZ further potentiated the inhibition of Glo1-specific activity (Figure 7c), MG-H1 accumulation (Figure 7d). OPN increased expression (Figure 7e) as well as migration (Figure 7f), invasion (Figure 7g) and proliferation (Figure 7h), thus further confirming our mechanistic results.

## 3. Discussion

This study aimed to investigate the role of ACh in promoting the migration, invasion and proliferation of PCa cells, specifically exploring the involvement of the Glo1/MG-H1 axis with the involvement of OPN. Our working hypothesis was that ACh would enhance these biological responses in PCa cells through the down-regulation of Glo1 and subsequent accumulation of MG-H1, which would activate downstream pro-tumorigenic pathways, including the up-regulation of OPN. The results of our experiments confirm this hypothesis in LNCaP cells, where ACh induced significant changes in Glo1 activity, MG-H1 levels, and OPN expression, while suggesting that a separate mechanism of action of ACh, yet to be discovered, might occur in PC3 cells, leading to the same increase in cell migration, invasion and proliferation. Our findings suggest that the Glo1/MG-H1 axis and OPN may play a crucial role in ACh-driven metastatic behavior in a subset of PCa, that is, the less aggressive ones.

By providing evidence on a novel mechanism through which ACh promotes the migration and invasion of LNCaP PCa cells, we confirm the emerging pro-tumorigenic role of this molecule, expanding our knowledge of the mechanisms by which ACh plays such a role in PCa cells and our understanding of cholinergic signaling in cancer biology.

The pro-migratory and pro-invasive effects of ACh observed in our study align with previous findings that cholinergic signaling plays a key role in cancer progression. Cholinergic signaling, through the activation of muscarinic acetylcholine receptors (mAChRs), has been implicated in the regulation of various cancer cell behaviors, including migration, invasion and metastasis. Several studies have demonstrated that ACh can stimulate cell motility in different cancer types, including breast [23,24], colon [25] and lung [26,27] cancers, by activating signaling pathways that involve integrins, extracellular matrix remodeling and cytoskeletal rearrangements. ACh also contributes to sustaining PCa progression by promoting cell proliferation, migration and invasion [3,4]. Our results further extend these findings. Interestingly, our data show that ACh has a more pronounced effect on less aggressive LNCaP than on metastatic PC3 cells. This supports the hypothesis that cholinergic signaling may play a more significant role in the early stages of PCa progression or in the context of cells with lower intrinsic metastatic potential. This differential response may be due to the distinct molecular characteristics of the two cell lines, such as the different responsiveness to androgens (androgen-sensitive LNCaP cells versus androgen-independent PC3 cells) [28], or it can be hypothesized that the more glycolytic phenotype typical of metastatic PC3 cells [29,30] could generate a higher intracellular amount of MG than in less aggressive LNCaP cells, thus making the former cells insensitive to Ach’s effects on Glo1. Indeed, future studies are needed to explore the mechanistic differences that underpin this specificity.

Importantly, our study introduces a novel molecular mechanism by which ACh promotes cell migration and invasion in LNCaP cells. We observed that ACh treatment led to a significant down-regulation of Glo1 and an accumulation of MG-H1.

Previous research has established that elevated levels of MG and the consequent accumulation of MG-H1 are associated with tumorigenesis, metastasis and poor prognosis [9,31]. In this context, our results are consistent with these findings, suggesting that ACh-induced Glo1 down-regulation and MG-H1 accumulation may promote cancer cell migration and invasion, as well as proliferation [32], by inducing a pro-tumorigenic environment.

Interestingly, our results highlight the differential response of PCa cell lines to ACh, with LNCaP cells (a less aggressive, androgen-sensitive model) showing a stronger migratory and invasive, as well as proliferative, response compared to PC3 cells (a more aggressive, androgen-independent model). This raises the possibility that cholinergic signaling could have distinct roles in different stages of cancer progression, potentially exerting more pronounced effects during early or less aggressive stages.

The up-regulation of OPN following ACh treatment in LNCaP cells further corroborates the role of this signaling pathway in cancer progression. OPN, a multifunctional extracellular matrix protein, has been extensively linked to cancer progression due to its role in promoting cell adhesion, migration and invasion, with elevated OPN levels being associated with poor prognosis in various cancers, including PCa [12,13,14,33]. Our results extend this body of literature by showing that ACh-induced OPN expression is associated with Glo1 down-regulation and MG-H1 accumulation. In line with previous studies, our data suggest that OPN may act as a downstream effector in the ACh-driven migration, invasion and proliferation observed in LNCaP cells [12,13,14]. This underscores the potential of targeting OPN as a therapeutic strategy to limit tumor spread in cancers driven by aberrant cholinergic signaling. Although silencing the OPN gene would further validate its direct role in driving the observed biological responses, the current data, including upstream perturbation experiments (carnosine, DNPZ) and consistent phenotypic results, provide strong support for the proposed model.

It is well known that MG-H1 can induce oxidative stress and act as secondary source of ROS [34]. Moreover, MG-H1 exerts its effect mostly via activation of RAGE, a cell surface receptor that initiates multiple intracellular signaling pathways, favoring a pro-oxidant environment through NADPH oxidase activation and the generation of high levels of ROS [35]. Interestingly, there is also evidence that ROS, particularly H_2_O_2_, can induce OPN expression [36], and that, while direct evidence linking ROS-induced OPN expression in PCa is limited, studies in another cancer type indicate that OPN can modulate ROS levels through specific signaling pathways [37]. These findings suggest that similar mechanisms might be at play in PCa, warranting further investigation to elucidate the role of ROS in OPN-related tumor progression through the Glo1/MG-H1 axis.

Our findings that carnosine, a scavenger of MG, abrogates ACh-induced OPN up-regulation and restores migration and invasion in LNCaP cells suggest that targeting the Glo1/MG-H1 axis and OPN could be a promising therapeutic strategy in PCa. This is consistent with previous studies showing that MG scavengers can reduce the pro-tumorigenic effects of MG and its derived AGEs [38,39]. In addition, our use of donepezil (DNPZ), an acetylcholinesterase (AChE) inhibitor whose efficacy has been demonstrated both through AChE activity (Figure 7a,b) and ACh accumulation detection (Appendix A), further supports the idea that ACh-induced effects on Glo1, MG-H1 and OPN are key mediators of the observed pro-tumorigenic effects. These results emphasize the potential of modulating cholinergic signaling and its downstream molecular pathways to inhibit cancer cell migration, invasion and proliferation, particularly in less aggressive prostate cancer models such as LNCaP. In light of the findings from this study, our data underscore the potential of cholinergic signaling as a driver of PCa metastasis. However, the differential response between LNCaP and PC3 cells to Ach—in particular, the involvement of the Glo1/MG-H1 axis and OPN only in LNCaP cells—could provide insights into the biological behaviour related to the progression of early stages of PCa. Moreover, interventions targeting the Glo1/MG-H1 axis and OPN might need to be tailored to the specific characteristics of each tumor subtype.

It is important to point out that our results, based on in vitro models, can be helpful in opening new avenues for in vivo investigation into the physiological relevance of ACh-driven Glo1/MG-H1 axis and OPN in PCa progression and the potential use of carnosine as a therapeutic agent. Moreover, it is well established that ACh exerts its effects through the activation of both nicotinic (nAChRs) and muscarinic (mAChRs) acetylcholine receptors [40]. Notably, the use of specific antagonists targeting these receptor subtypes warrants further investigation, as it may aid in elucidating the contribution of ACh signaling to the responses observed in our study, an approach that should be considered in future research.

Finally, future studies could also investigate whether the Glo1/MG-H1 axis and OPN are similarly involved in ACh-driven metastasis in other cancer types and whether interventions targeting this pathway could be clinically relevant in reducing tumor progression and metastasis. Certainly, xenograft models and RNA sequencing experiments would not only provide further validation of the role of ACh in promoting cell proliferation, migration and invasion—hallmarks of tumor progression—but also offer deeper insights into additional molecular pathways mediating these processes. Such findings could significantly enhance our understanding of ACh-driven oncogenic mechanisms and open promising avenues for future research.

## 4. Materials and Methods

### 4.1. Reagents

Acetylcholine (ACh) (cat. A6625), L-carnosine (cat. C9625), donepezil (DNPZ) (cat. D6821) and trypan blue (cat. T8154) were purchased from Merck Spa (Milan, Italy).

### 4.2. Cell Models and Culture Conditions

LNCaP and PC3 prostate cancer cell lines were purchased from the American Type Culture Collection (ATCC) and cultured in RPMI medium at 37 °C and 5% CO_2_ as per the supplier’s recommendations [9,31].

### 4.3. RNA Isolation, Reverse Transcription and Real-Time Reverse Transcriptase-Polymerase Chain Reaction (RT-PCR)

Total RNA was extracted using TRIzol Reagent (ThermoFisher Scientific, Milan, Italy). Subsequently, 1 µg of RNA was used to synthesise DNA with the RevertAid™ H Minus First Strand cDNA Synthesis Kit (ThermoFisher Scientific, Milan, Italy). Finally, the expression levels of Glo1 and OPN (in relation to β-actin) were evaluated by RT-PCR using the MX3000P Real-Time PCR System (Agilent Technology, Milan, Italy).The sequences of primers were as follows: Glo1, sense 5′-CTCTCCAGAAAAGCTACACTTTGAG-3′ and antisense 5′-CGAGGGTCTGAATTGCCATTG-3′; OPN, sense 5′-GCCGAGGTGATAGTGTGGTT-3′ and antisense 5′-TGAGGTGATGTCCTCGTCTG-3′; β-actin, sense 5′-CACTCTTCCAGCCTTCCTTCC-3′ and antisense 5′-ACAGCACTGTGTTGGCGTAC-3′. The PCR reactions were performed in a total volume of 20 µL that contained 25 ng of cDNA, 1× Brilliant II SYBR^®^ Green QPCR Master Mix, ROX as Reference Dye and 600 nM of specific primers. The thermal cycling conditions comprised the following: initial denaturation at 95 °C for 5 min (1 cycle), denaturation at 95 °C for 20 s, and annealing at 60 °C for 30 s (45 cycles). To verify the potential co-amplification of unspecific targets, the melting curves for each primer pair were performed under standard conditions. The 2^−(ΔΔCT)^ method was used to perform a comparative analysis of gene expression [41].

### 4.4. Cell Lysates

In order to lyse sub-confluent cells, the pre-cooled radioimmunoprecipitation assay (RIPA) lysis buffer, which was supplied by ThermoFisher Scientific (Milan, Italy), was utilised. The Halt Protease Inhibitor Cocktail and Halt Phosphatase Inhibitor Cocktail, which were also supplied by ThermoFisher Scientific (Milan, Italy), were included as per the manufacturer’s instructions. The protein extracts obtained were utilised for the measurement of enzyme activity and for the quantification of total protein content by means of the Lowry et al. method [42], with bovine serum albumin as a standard.

### 4.5. Glo1 Enzyme Activity

The activity of Glo1 was assessed in accordance with the established protocol [31,41]. The assay solution comprised 0.1 mol/L sodium phosphate buffer (pH 7.2), 2 mmol/L MG and 1 mmol/L GSH. The reaction was monitored spectrophotometrically by following the increase in light absorption at 240 nm and 25 °C. The activity of Glo1 was defined as 1 μmol of S-d-lactoylglutathione produced per minute, representing one unit of activity.

### 4.6. Western Blot

Western blot was performed as described previously [41]. Briefly, samples were boiled in Laemmli buffer, resolved by SDS-PAGE and blotted onto a nitrocellulose membrane. Roti-Block (room temperature, RT, 1 h) was used to block non-specific binding sites. Membranes were incubated with the appropriate primary Abs: mouse anti-Glo1 mAb (dilution 1:1000, Santa Cruz, cat. sc-133144, DBA Italia S.r.l., Milan, Italy), mouse anti-OPN mAb (dilution 1:1000, Santa Cruz, cat. sc-73631, DBA Italia S.r.l., Milan, Italy) and mouse anti-β-actin mAb (dilution 1:1000, Santa Cruz, cat. sc-517582, DBA Italia S.r.l., Milan, Italy) as an internal loading control was performed overnight at 4 °C. The membranes were then incubated with the appropriate HRP-conjugated secondary Ab (RT, 1 h), and ECL was used as the detection system (Amersham Pharmacia, Milan, Italy).

### 4.7. MG-H1 Detection

The OxiSelect Methylglyoxal Competitive ELISA kit (DBA Italia S.r.l., Milan, Italy) was utilized in accordance with the manufacturer’s instructions to measure the levels of MG-H1 [43,44].

### 4.8. Migration and Invasion

Migration was evaluated by the CytoSelect 24-Well Cell Migration Assay kit (cat. CBA-100-5, DBA Italia S.r.l., Milan, Italy) [9] and invasion by the CytoSelect 24-Well Cell Invasion Assay kit (cat. CBA-110, DBA Italia S.r.l., Milan, Italy) [9], both according to the manufacturer’s instructions.

### 4.9. Cell Proliferation

Cell proliferation was evaluated by the trypan blue exclusion assay, with viable cells counted using a hemocytometer under a light microscope [45].

### 4.10. ACh Levels

ACh levels were measured using the Human Acetylcholine ELISA Kit (cat. ab287811, Prodotti Gianni, Milan, Italy) according to the manufacturer’s instructions and expressed as µmol/mg protein [46].

### 4.11. AChE Enzyme Activity

AChE activity was evaluated at 20 °C following a variant of the spectrophotometric method of Ellman et al. [47], using acetylthiocholine (ATC) as the substrate [48]. Briefly, the assay mixture was composed of 0.89 mL of 0.1 M Na-phosphate buffer, pH 7.2, that contained 0.5 mM DTNB, 0.1 mL of 1.0 mM substrate and 0.01 mL of cell lysate. The product of thiocholine reactions with DTNB was spectrophotometrically determined at 412 nm (ξ = 13,600). In addition, AChE activity was evaluated by a non-denaturing PAGE-based method as previously described [48], loading the same amount of total proteins. Staining for AChE activity was achieved according to Karnovsky and Roots, using 3 mM ATC as substrate [48]. Two independent experiments were conducted (Appendix A).

### 4.12. Statistical Analysis

The results were analyzed by using GraphPad Prism 9.0.2 software and expressed as the means ± standard deviation (SD) of three independent experiments. One-way analysis of variance with correction for multiple comparisons was employed to determine differences among groups. Statistical significance was set at *p* < 0.05.

## 5. Conclusions

In summary, our study demonstrates that ACh promotes migration, invasion and proliferation in LNCaP PCa cells via a Glo1-dependent, MG-H1-mediated up-regulation of OPN (Figure 8).

These findings provide new insights into the molecular mechanisms underlying ACh-driven metastasis and suggest that targeting the Glo1/MG-H1 pathway and OPN could offer a novel strategy to interfere with the progression of prostate cancers where cholinergic signaling plays a pivotal role before it becomes clinically aggressive and potentially uncurable.

Our study also highlights the importance of understanding the differential responses of cancer cell lines to ACh, which could have implications for personalized therapeutic strategies targeting cholinergic signaling in cancer.

However, given the complexity of cholinergic signaling in cancer and its potential interaction with various signaling pathways, future research should aim to address several key questions. First, the precise mechanisms by which ACh regulates Glo1 and MG-H1 in cancer cells need to be explored in more detail. On the basis of the results reported here, a direct effect of ACh on Glo1 appears unlikely because Glo1 mRNA level and activity, both significantly reduced in LNCaP cells, were not affected by ACh treatment in PC3 cells, thus suggesting the involvement of a LNCaP cell-specific upstream pathway in the mechanism of action of ACh. Understanding whether ACh directly modulates Glo1 expression or whether it activates upstream pathways involved in oxidative stress or inflammation could provide valuable insights into this process.

Second, the role of cholinergic signaling in aggressive cancer models, such as PC3 cells, should be investigated further to determine if the observed differential response between LNCaP and PC3 cells is due to intrinsic differences in receptor expression, signaling networks, or other factors such as the tumor microenvironment. This will help clarify the broader applicability of cholinergic-targeted therapies across various prostate cancer subtypes.

## Figures and Tables

**Figure 1 ijms-26-04107-f001:**
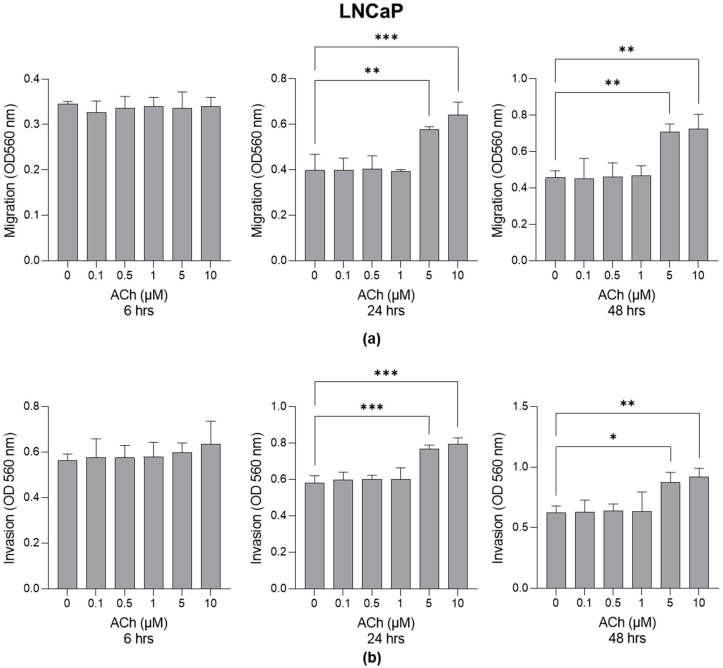
Effect of Acetylcholine (ACh) on LNCaP cell migration and invasion. (**a**) LNCaP migration and (**b**) invasion. The cell migrative and invasive capacities were measured by specific assays as described in Section 4. The histograms indicate mean ± SD of three separate independent experiments. * *p* < 0.05; ** *p* < 0.01; *** *p* < 0.001.

**Figure 2 ijms-26-04107-f002:**
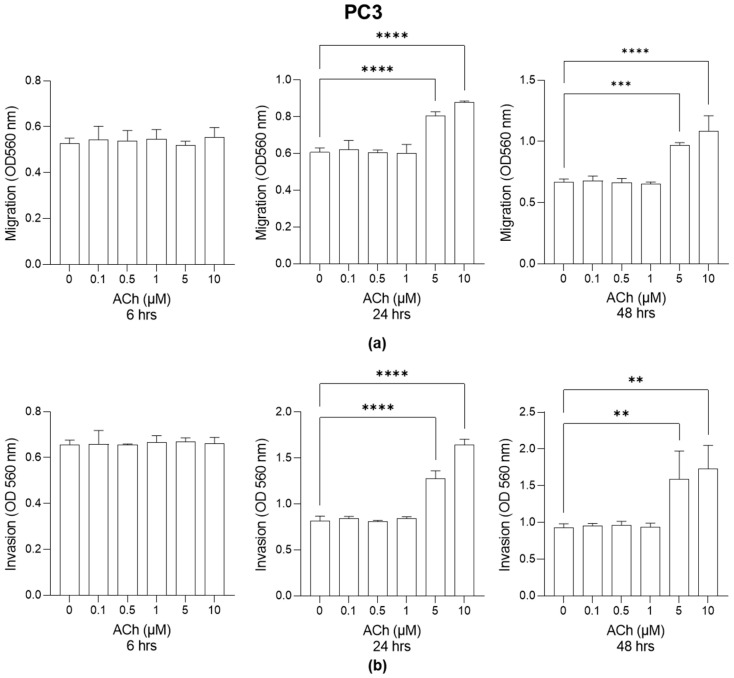
Effect of Acetylcholine (ACh) on PC3 cell migration and invasion. (**a**) PC3 migration and (**b**) invasion. The cell migrative and invasive capacities were measured by specific assays as described in Section 4. The histograms indicate mean ± SD of three separate independent experiments. ** *p* < 0.01; *** *p* < 0.001; **** *p* < 0.0001.

**Figure 3 ijms-26-04107-f003:**
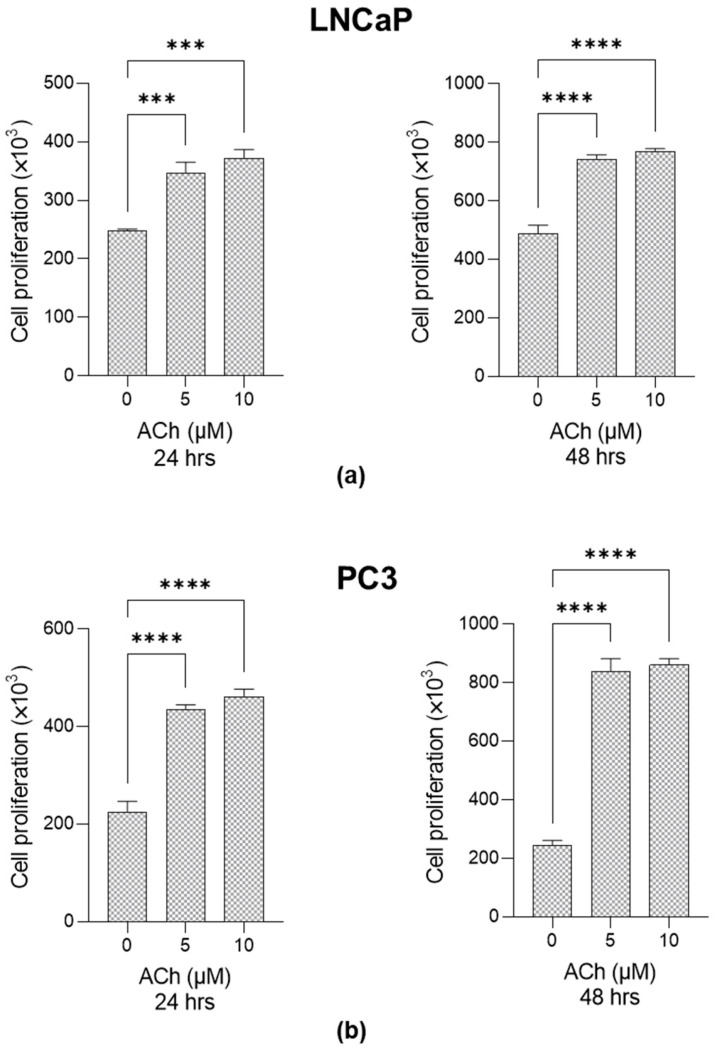
Effect of Acetylcholine (ACh) on LNCaP and PC3 cell proliferation. (**a**) LNCaP and (**b**) PC3 cells were exposed to 5 µM and 10 µM Ach, and cell proliferation was evaluated at 24 and 48 h post-treatment by trypan blue exclusion assay. The histograms indicate mean ± SD of three separate independent experiments. *** *p* < 0.001; **** *p* < 0.0001.

**Figure 4 ijms-26-04107-f004:**
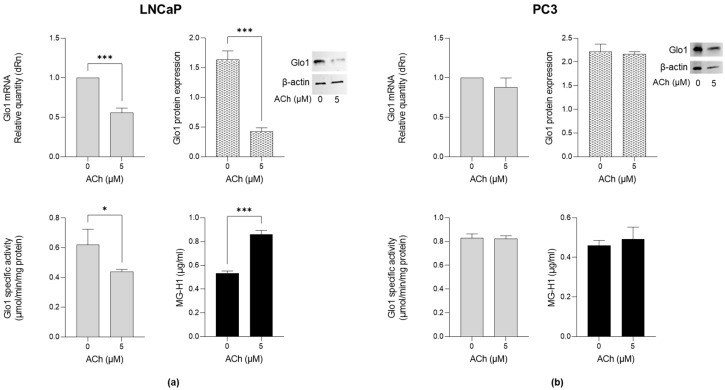
Effects of ACh on Glo1 expression and MG-H1 levels in LNCaP and PC3 cells. (**a**) LNCaP and (**b**) PC3 cells were treated with 5 µM ACh for 24 h, and Glo1 mRNA and protein expression or specific activity, as well as MG-H1 levels, were measured by RT-qPCR, western blotting, spectrophotometry, and a specific ELISA kit, respectively, as described in the Materials and Methods. The histograms indicate mean ± SD of three separate independent experiments. * *p* < 0.05; *** *p* < 0.001.

**Figure 5 ijms-26-04107-f005:**
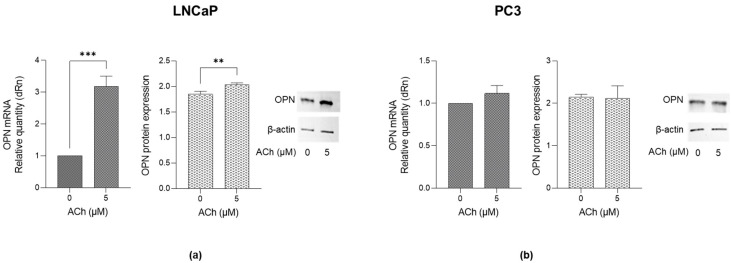
Effect of ACh on OPN expression in LNCaP and PC3 cells. (**a**) LNCaP and (**b**) PC3 cells were treated with 5 µM ACh for 24 h, and OPN mRNA and protein expression were measured by RT-qPCR and western blotting, respectively. The histograms indicate mean ± SD of three separate independent experiments. ** *p* < 0.01, *** *p* < 0.001.

**Figure 6 ijms-26-04107-f006:**
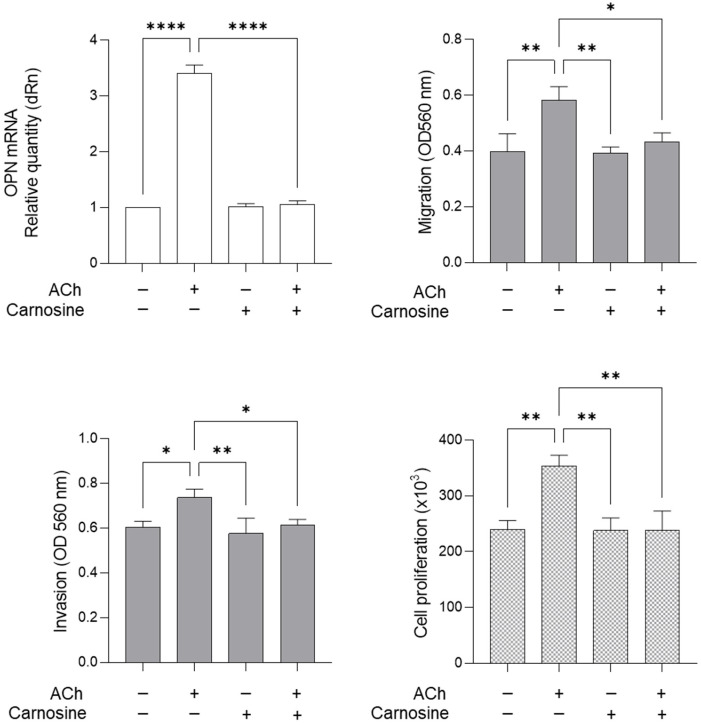
Effect of carnosine on OPN expression, migration, invasion and proliferation in LNCaP cells. LNCaP cells were pre-treated with 10 mM carnosine for 48 h and then exposed to 5 µM ACh for 24 h; OPN mRNA was measured by RT-qPCR, migration and invasion by specific assays, and proliferation was measured by trypan blue exclusion assay, as described in Section 4. The histograms indicate mean ± SD of three separate independent experiments. * *p* < 0.05; ** *p* < 0.01; **** *p* < 0.0001.

**Figure 7 ijms-26-04107-f007:**
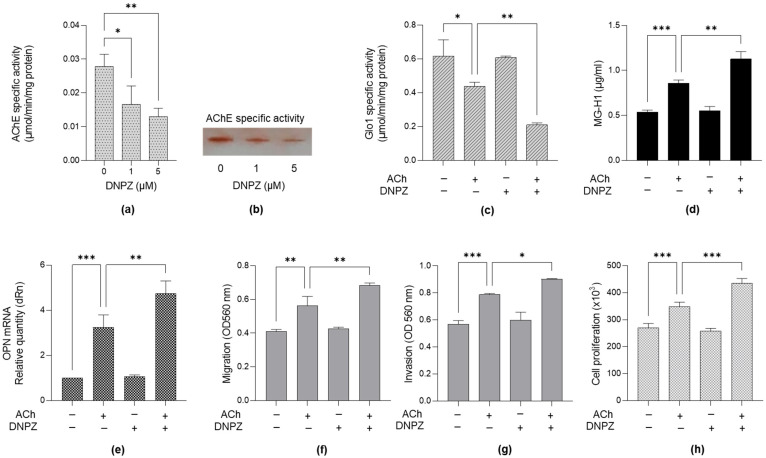
The effects of Donepezil (DNPZ) on Glo1-specific activity, MG-H1 levels, OPN expression, migration, invasion and proliferation in LNCaP cells. The inhibitory action of DNPZ on Acetylcholinesterase (AChE) was measured using either (**a**) a specific enzymatic assay by spectrophotometry or (**b**) electrophoresis, as described in Section 4. LNCaP cells were pre-treated with 5 µM DNPZ for 24 h and then exposed to 5 µM Acetylcholine (ACh) for an additional 24 h. (**c**) Glo1-specific activity, (**d**) MG-H1 levels and (**e**) OPN mRNA expression, as well as (**f**) migration, (**g**) invasion and (**h**) proliferation, were measured by spectrophotometry, a specific ELISA kit, RT-qPCR and specific assays or trypan blue exclusion assay, respectively, as described in Materials and Methods. The histograms represent the mean ± SD of three independent experiments. * *p* < 0.05; ** *p* < 0.01; *** *p* < 0.001.

**Figure 8 ijms-26-04107-f008:**
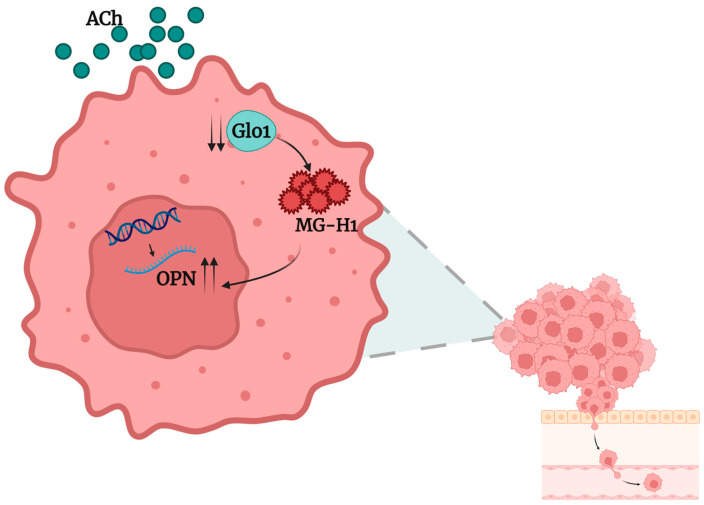
Acetylcholine (ACh) sustains LNCaP prostate cancer cell migration, invasion and proliferation through the glyoxalase 1 (Glo1)/MG-H1 axis with the involvement of osteopontin (OPN), whose transcription levels are induced (↑) by the intracellular accumulation of MG-H1 consequent to glyoxalase 1 (Glo1)-decreased (↓) enzyme-specific activity. Created in BioRender, https://app.biorender.com (accessed on 1 January 2025).

## Data Availability

The original contributions presented in the study are included in the article/Appendix A. Further inquiries can be directed to the corresponding author.

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
