# Peer review of "Acetylcholine Sustains LNCaP Prostate Cancer Cell Migration, Invasion and Proliferation Through Glyoxalase 1/MG-H1 Axis with the Involvement of Osteopontin"

_ijms, 2025, doi:10.3390/ijms26094107_

Round 1

Reviewer 1 Report

Comments and Suggestions for Authors

  1. for the figure6, you should modify the figure legand including the information for abcdefg panels.
  2. for figure 1 and 2, you only provide the invasion and migration data for PC3 and LNCap. you may consider the proliferation assay or an in vivo xenograft study to check the bone metastasis.
  3. for figure 3, 4,5, 6 figures, you only showed the mRNA change under treatment. Please provide the protein expression level change under same treatment.
  4. I suggest to conduct a RNA-seq under ACh treatment to provide more comprehensive understanding for the effect of ACh for transcriptome. You can also find a new gene regulated by ACh to continue study the mechanisms. 

Author Response

Dear Reviewer,

Thank you very much for providing helpful comments and suggestions aimed at improving the manuscript and strengthen the impact of our research work.

Detailed responses addressing point-by-point the issues raised in the individual comments are provided below.

Looking forward to seeing your response,

Yours sincerely,

Prof. Cinzia Antognelli

______________________________________

Department of Medicine and Surgery

University of Perugia

  1. Severi Square, 1

06129 Sant’Andrea delle Fratte (PG), Italy

Phone: +39-075.5858354

e-mail: cinzia.antognelli@unipg.it

Comment 1: “for the figure6, you should modify the figure legand including the information for abcdefg panels.”

Authors’ response: Thank you for pointing this out. We have now modified the legend to Figure 6 (now Figure 7) as suggested by the Reviewer.

Comment 2: “for figure 1 and 2, you only provide the invasion and migration data for PC3 and LNCap. You may consider the proliferation assay or an in vivo xenograft study to check the bone metastasis.”

Authors’ response: Thank you for your valuable suggestions. We have now assessed the effect of ACh on the proliferation of LNCaP and PC3 cells at concentrations of 5 and 10 µM, with treatments lasting 24 and 48 hours. Consistently with the literature, we observed that ACh promotes cell proliferation in both LNCaP and PC3 cells. Additionally, our findings indicate that Glo1/MG-H1-OPN not only regulates migration and invasion but also contributes to proliferation. Hence, the title, M&M, Results (the new paragraph “2.2. Effect of ACh on LNCaP and PC3 cell proliferation” was added), Figures and Discussion were modified accordingly.

As to the alternative suggestion by the Reviewer, we fully agree with him/her. However, while conducting an in vivo xenograft study to investigate bone metastasis is undoubtedly crucial, it is important to note that such an experiment would require a significant amount of time and a collaboration with other research groups, since our research group does not have access or expertise to manage an animal facility, and, even as part of a collaboration with others colleagues, Italian regulations in this ambit would require a detailed project proposal along with ethical approval from the animal ethics committee, which would take a very long time. We will include a comment in the discussion to highlight this aspect as a future direction for research.

Comment 3: “for figure 3,4,5,6 figures, you only showed the mRNA change under treatment. Please provide the protein expression level change under same treatment.”

Authors’ response: We appreciate the reviewer’s suggestion to assess the protein expression of Glo1 and OPN, and indeed, we have now assessed the protein expression level of Glo1 and OPN upon 5 µM ACh treatment and control (please, see Figure 4 and 5 of the revised manuscript, originally Figure 3 and 4). The results showed a consistent and overlapping trend across all three levels—transcriptional, translational, and functional—indicating a good correlation among them. Based on this concordance, we considered it redundant to repeat protein expression analysis in the subsequent experiments, as it would not provide additional mechanistic insights beyond what was already established. A similar observation was made for OPN, whose expression pattern also followed that of its corresponding mRNA (please, see Figure 5 of the revised manuscript). Nevertheless, we remain open to including protein expression data if the reviewer deems it essential for the completeness of the study.

Comment 4: “I suggest to conduct a RNA-seq under ACh treatment to provide more comprehensive understanding for the effect of ACh for transcriptome. You can also find a new gene regulated by ACh to continue study the mechanism”

Authors’ response: Thank you also for this additional suggestion. We fully agree that RNA-seq analysis would provide valuable and comprehensive insights into the effects of ACh on the transcriptome and this approach represents a valuable continuation of our work and will be considered in future studies to further elucidate the broader molecular pathways modulated by ACh in the tumor microenvironment. However, the primary objective of our study was to specifically investigate the involvement of Glo1/MG-H1 axis and OPN in the pro-tumorigenic role of ACh. While transcriptomic profiling represents an important direction for future research, our current work was focused on elucidating this targeted mechanistic link.

Reviewer 2 Report

Comments and Suggestions for Authors

[Major Issues]

Comment 1: Clarification of ACh-driven mechanisms and experimental validation

The manuscript suggests that acetylcholine downregulates glyoxalase 1, leading to increased MG-H1 accumulation and osteopontin upregulation. Additionally, Donepezil, an acetylcholinesterase inhibitor, was used to enhance ACh levels and evaluate its effects on the Glo1/MG-H1/OPN axis. However, there are some concerns that should be explained in the manuscript.

1) According to the previous studies (https://doi.org/10.1002/advs.202304079), In PC12 cancer cells, an increase in ROS has been observed upon ACh treatment. Considering that this study is related to ACh utilization, could this be linked to mitochondrial ROS? Given that carnosine has antioxidant properties and removes reactive oxygen species, I believe this is worth considering. In addition to the data presented in the manuscript, please explain the impact of the Glo1/MG-H1 axis in relation to ROS levels.

2) Based on the Reviewer’s opinion, how the intracellular concentration of Ach changes upon treatment with the acetylcholinesterase inhibitor, donepezil. This data would support the increase in OPN mRNA level as well as the enhancement of migration and invasion observed with the simultaneous treatment of ACh and DNPZ.

3) Additionally, please explain that the effect of alternative AChE inhibitor and that a muscarinic or nicotinic receptor antagonist could help delineate the specificity of donepezil’s effect.

4) The Reviewer recommends conducting an OPN knockdown experiment to verify whether the reduction in OPN expression leads to decreased migration and invasion. The results would strengthen the credibility of the Glo1/MG-H1/osteopontin axis mechanism proposed in your study.

5) Further, in vivo validation would strengthen these findings. This study relies on in vitro models using LNCaP and PC3 cells. If the physiological relevance of ACh levels in the tumor microenvironment is examined, it could provide valuable insights. Additionally, such validation could highlight the potential of carnosine as a therapeutic agent.

[Minor Issues]

  1. In Results and Discussion, 3.1 & 3.2 have same title.

Author Response

Dear Reviewer,

Thank you very much for providing helpful comments and suggestions aimed at improving the manuscript and strengthen the impact of our research work.

Detailed responses addressing point-by-point the issues raised in the individual comments are provided below.

Looking forward to seeing your response,

Yours sincerely,

Prof. Cinzia Antognelli

______________________________________

Department of Medicine and Surgery

University of Perugia

  1. Severi Square, 1

06129 Sant’Andrea delle Fratte (PG), Italy

Phone: +39-075.5858354

e-mail: cinzia.antognelli@unipg.it

Major issues

Comment 1: “According to the previous studies (https://doi.org/10.1002/advs.202304079), In PC12 cancer cells, an increase in ROS has been observed upon ACh treatment. Considering that this study is related to ACh utilization, could this be linked to mitochondrial ROS? Given that carnosine has antioxidant properties and removes reactive oxygen species, I believe this is worth considering. In addition to the data presented in the manuscript, please explain the impact of the Glo1/MG-H1 axis in relation to ROS levels.”

Authors’ response: We thank the Reviewer for his/her suggestion. However, the link provided (DOI: 10.1002/advs.202304079) appears to refer to an article that, upon careful reading, does not seem to correspond to the content or context indicated in the reviewer’s comment [The article is titled "A Bioinspired Flexible Sensor for Electrochemical Probing of Dynamic Redox Disequilibrium in Cancer Cells" by Zhongyuan Zeng et al., is published in Advanced Science, and presents a biologically inspired flexible sensor for the electrochemical monitoring of dynamic redox disequilibrium in cancer cells. The developed sensor, PtNFs/CoPi@CC, is designed to detect in real time the production of hydrogen peroxide (Hâ‚‚Oâ‚‚) within cells, a key indicator of oxidative stress induced by ascorbic acid (AA) and its oxidized form, dehydroascorbic acid (DHA)]. We apologize if this is an oversight on our part and kindly ask the reviewer to provide additional details to help us locate the correct article. Moreover, to the best of our knowledge, PC12 cells are not classified as cancer cells. Finally, based on our review of the literature, there are no direct studies demonstrating that ACh induces mitochondrial ROS production specifically in prostate cancer cells. However, some studies have examined the effects of ACh on cellular mechanisms that could indirectly influence ROS levels. For example, one study explored the effects of ACh on H9c2 cardiac cells under hypoxia/reperfusion-induced oxidative stress, where ACh increased ATP synthesis and mitochondrial DNA content, suggesting, however, a potential protective role against oxidative stress, and no direct effects on mitochondrial ROS production were observed in this model (PubMed ID: 25402826). Additionally, another study investigated the activation of the muscarinic acetylcholine receptor M1 (CHRM1) in prostate cancer cells, showing that CHRM1 activation promoted cell migration and invasion through the Hedgehog signaling pathway. However, no direct effects on mitochondrial ROS production were reported in this context either (PubMed ID: 30027929). Therefore, while ACh has been shown to affect various cellular parameters, there is currently no direct evidence indicating that ACh induces mitochondrial ROS production in prostate cancer cells. To provide a more appropriate and accurate response, we kindly request additional information or clarification regarding the specific sources or studies the Reviewer refers to. This will help us address the comment more thoroughly and ensure that our response is as precise as possible.

Regarding the impact of the Glo1/MG-H1 axis in relation to ROS levels, this is indeed a very interesting aspect. It is well known, in fact, that MG-H1 can increase oxidative stress and may act as secondary sources of ROS (PMID: 39019314). Moreover, it is known that MG-H1 can exert its effect mostly via activation of RAGE, a cell surface receptor that initiates multiple intracellular signaling pathways, favoring a pro-oxidant environment through NADPH oxidase activation and generation of high levels of ROS (PMID: 38790901).

More interestingly, there is also evidence that ROS, particularly Hâ‚‚Oâ‚‚, can induce OPN expression. A study published in the Journal of Biological Chemistry demonstrated that Hâ‚‚Oâ‚‚ stimulates OPN expression through both transcriptional and translational mechanisms (PMID: 24247243), and, while direct evidence linking ROS-induced OPN expression in prostate cancer is limited, studies in another cancer type indicate that OPN can modulate ROS levels through specific signaling pathways (PMID: 35418176). These findings suggest that similar mechanisms might be at play in prostate cancer, warranting further investigation to elucidate the role of ROS in OPN-related tumor progression through Glo1/MG-H1 axis.

In particular, based on the above considerations, in the present study, ROS could indeed participate to OPN-mediated proliferation/migration/invasion via MG-H1 accumulation, consequent to Glo1 reduced functionality, driven by ACh. Understanding this could provide valuable insights into the way how Glo1/MG-H1 axis trigger OPN-dependent LNCaP invasion, migration and proliferation, which will be object of future perspectives.

We have added a comment in discussion.

Comment 2: “Based on the Reviewer’s opinion, how the intracellular concentration of Ach changes upon treatment with the acetylcholinesterase inhibitor, donepezil. This data would support the increase in OPN mRNA level as well as the enhancement of migration and invasion observed with the simultaneous treatment of ACh and DNPZ.

Authors’ response: Thank you very much for your suggestion. We have now measured ACh levels upon treatment with ACh, DNPZ and a combination of both. Results are shown in Figure S1.

Comment 3: “Additionally, please explain that the effect of alternative AChE inhibitor and that a muscarinic or nicotinic receptor antagonist could help delineate the specificity of donepezil’s effect.”

Authors’ response: Thank you for your suggestion. We have shown that donepezil effectively inhibits AChE, as indicated by a measurable reduction in its enzymatic activity and ACh accumulation, which is associated with specific biological responses (proliferation, migration, invasion). Given these findings, the mechanism of action appears to be consistent with the known pharmacological profile of donepezil as an AChE inhibitor. In this context, we think that the use of additional AChE inhibitors may not be strictly necessary to confirm specificity, as the biochemical and functional results observed are consistent with the expected effects of AChE inhibition. Conversely, since it is well established that ACh can exert its effects through the activation of both nicotinic (nAChRs) and muscarinic (mAChRs) acetylcholine receptors, the use of specific antagonists targeting these receptor subtypes could be very interesting and certainly warrants further investigation, as it may aid in elucidating the contribution of ACh signaling to the responses observed in our study, an approach that will be considered in future research. We have included a comment about this in the Discussion.

Comment 4: “The Reviewer recommends conducting an OPN knockdown experiment to verify whether the reduction in OPN expression leads to decreased migration and invasion. The results would strengthen the credibility of the Glo1/MG-H1/osteopontin axis mechanism proposed in your study.”

Authors’ response: Thank you for your suggestion of silencing the OPN gene to further validate its role in driving the observed biological responses. We fully agree that such an experiment would strengthen our proposed mechanistic model. However, despite several attempts, we have not yet been able to achieve efficient and reproducible OPN silencing. We have tested different siRNAs and transfection conditions, but knockdown efficiency remains suboptimal and inconsistent across biological replicates. Given the relatively high basal expression of the gene and its potential involvement in essential cellular functions, we suspect that compensatory mechanisms may be contributing to the technical challenges we have encountered. We may consider alternative strategies to address this limitation in future experiments. While we acknowledge the importance of this validation step, we believe that the current data, including upstream perturbation experiments (carnosine, DNPZ) and consistent phenotypic results, provide strong support for the proposed model. We have now added a comment on this in the Discussion and “mitigated”, wherever possible, the description of Glo1/MG-H1-OPN axis.

Comment 5: “Further, in vivo validation would strengthen these findings. This study relies on in vitro models using LNCaP and PC3 cells. If the physiological relevance of ACh levels in the tumor microenvironment is examined, it could provide valuable insights. Additionally, such validation could highlight the potential of carnosine as a therapeutic agent.”

Authors’ response: Thank you for your valuable comment. Indeed, it is important to point out that our results based on in vitro models may be helpful to open new avenues for in vivo investigation of the physiological relevance of the ACh-driven Glo1/MG-H1 axis and OPN in PCa progression and the potential use of carnosine as a therapeutic agent, which will be further explored in future analyses. We have included this in the Discussion.

The performance of an in vivo study is undoubtedly crucial. However, it is important to note that such an experiment would require a considerable amount of time and collaboration with other research groups, as our research group does not have access or expertise to manage an animal facility, and even in the context of collaboration with other colleagues, Italian regulations in this area would require a detailed project proposal along with ethical approval from the animal ethics committee, which would take a very long time indeed. We will include a comment in the discussion to highlight this aspect as a future research direction.

[Minor Issues]

Comment: 1. In Results and Discussion, 3.1 & 3.2 have same title.

Authors’ response: Thank you for your comment. However, we would like to point out that sections 3.1 and 3.2 do not exist in the manuscript. We have written sections 3 and 4 which have different titles:  Section 3 is titled "Discussion" and Section 4 is titled "Results". We would be grateful if you could clarify this. We remain at your disposal for further clarification.

Round 2

Reviewer 1 Report

Comments and Suggestions for Authors

  1. Please provide the xenograft animal study of LNCap or PC3 cell lines to further confirm the function of acetylcholine for the proliferation. 
  2. Bulk RNA seq in LNCAP or PC3 cell line treated with acetylcholine can provide more evidence for this finding. 

Author Response

Comment 1: Please provide the xenograft animal study of LNCap or PC3 cell lines to further confirm the function of acetylcholine for the proliferation. 

Authors’ response: Dear Reviewer, thank you for your feedback. As you suggested in your first comment as an alternative to xenografts (“...You may consider the proliferation assay or an in vivo xenograft study to check the bone metastasis”), we have now conducted cell proliferation assays, which we believe provide meaningful data consistent with the objectives of our study, namely to evaluate the involvement of glyoxalase 1/MG-H1 axis in ACh-driven LNCaP prostate cancer cell migration, invasion and, now, proliferation. Thank you again for this valuable suggestion that helped us to improve our study. As previously said, at present, we lack the expertise and resources to carry out xenograft experiments. We also consider highlighting that such a request would require a considerable commitment and that, at present, the focus of our work is based on the data already obtained, which we believe are sufficient to support the main conclusions.

Comment 2: Bulk RNA seq in LNCAP or PC3 cell line treated with acetylcholine can provide more evidence for this finding.

Authors’ response: Dear Reviewer, thank you also for this renewed suggestion. We appreciate your interest in further exploring the molecular mechanisms underlying our findings. However, we believe that performing RNA-seq at this stage would not add significant new insights to our current work. Our study has already demonstrated a clear mechanism, and we consider that additional transcriptomic data, requiring further extensive and highly costly experiments, might not substantially enhance the main conclusions we have presented.

We remain open to future studies that could incorporate such an approach, but for now, we believe our current data sufficiently support the main message of our work.

In conclusion, we agree that both experimental approaches could potentially enhance our study by providing additional insights and recognize the potential value of these approaches but we believe that they can be considered promising avenues for future research. We have now added in Discussion, last paragraph this comment: “Certainly, xenograft models and RNA sequencing experiments would not only provide further validation of the role of ACh in promoting cell proliferation, migration, and invasiveness—hallmarks of tumor progression—but also offer deeper insights into additional molecular pathways mediating these processes. Such findings could significantly enhance our understanding of ACh-driven oncogenic mechanisms and open promising avenues for future research”.

We appreciate your understanding and your constructive feedback.

Reviewer 2 Report

Comments and Suggestions for Authors

Thank you for the kind responses.

Author Response

Dear Reviewer, thanks to you for your valuable suggestions.

Round 3

Reviewer 1 Report

Comments and Suggestions for Authors

Please consider conducing more experiments in the future.